# Assessment of the drugability of initial malaria infection through miniaturized sporozoite assays and high-throughput screening

Marie Miglianico[1,5], Judith M. Bolscher[1,5], Martijn W. Vos[1], Karin J. M. Koolen[1], Marloes de Bruijni[1], Deeya S. Rajagopal[1], Emily Chen[2], Michael Kiczun[3], David Gray[3], Brice Campo [4], Robert W. Sauerwein[1] & Koen J. Dechering [1✉]

The sporozoite stages of malaria parasites are the primary cause of infection of the vertebrate host and are targeted by (experimental) vaccines. Yet, little is known about their susceptibility to chemical intervention. Phenotypic high-throughput screens have not been feasible due to a lack of in vitro systems. Here we tested 78 marketed and experimental antimalarial compounds in miniaturized assays addressing sporozoite viability, gliding motility, hepatocyte traversal, and intrahepatocytic schizogony. None potently interfered with sporozoite viability or motility but ten compounds acted at the level of schizogony with IC50s < 100 nM. To identify compounds directly targeting sporozoites, we screened 81,000 compounds from the Global Health Diversity and reFRAME libraries in a sporozoite viability assay using a parasite expressing a luciferase reporter driven by the circumsporozoite promoter. The ionophore gramicidin emerged as the single hit from this screening campaign. Its effect on sporozoite viability translated into reduced gliding motility and an inability of sporozoites to invade human primary hepatocytes and develop into hepatic schizonts. While providing proof of concept for a small molecule sporontocidal mode of action, our combined data indicate that liver schizogony is more accessible to chemical intervention by (candidate) antimalarials.

[1] TropIQ Health Sciences, Nijmegen, The Netherlands. [2] Calibr, a division of The Scripps Research Institute, La Jolla, California, United States of America. [3] Drug Discovery Unit, University of Dundee, Dundee, United Kingdom. [4] Medicines for Malaria Venture, Geneva, Switzerland. [5] These authors contributed equally: Marie Miglianico, Judith M. Bolscher. ✉email: k.dechering@tropiq.nl

The fight against malaria, the most important human parasitic disease[1], gained impressive ground between 2000 and 2015 with an estimated 40% drop in deaths related to *Plasmodium falciparum* infections[2]. More recently, however, the number of cases and deaths has increased[3]. Resistance to artemisinins, the first-line therapies recommended for the treatment of uncomplicated malaria, is rising[4]. To support malaria control and elimination, novel therapies are needed to counter the rise of resistant parasites and to protect humans from contracting the disease in the first place[5].

Human malaria infection starts with the injection of the parasites in the sporozoite stage through the bite of an infected *Anopheles* mosquito. The sporozoites travel through the skin and bloodstream to finally invade hepatocytes, traversing a variety of host cells along the way to overcome cellular barriers[6,7]. After maturation and egress from the hepatocytes, the parasites enter red blood cells and start an asexual cycle which is marked by host cell lysis and causes the clinical symptoms. Part of these blood-stage parasites differentiate into the sexual stages which are called gametocytes and can infect new mosquitoes through the bloodmeal. Malaria can be prevented by inhibiting the development of liver-stage parasites (causal prophylaxis) or by eliminating the first round of asexual blood-stage replication (suppressive prophylaxis)[8]. Candidate protective vaccines primarily target the infectious sporozoite, which provides a population bottleneck in the malaria life cycle as only a dozen sporozoites are sufficient to establish a human infection[9]. Therefore, this stage has been intensely investigated, either to identify potential antigens, such as the circumsporozoite protein (CSP), the predominant surface protein used in the RTS,S candidate vaccine, or to develop attenuated sporozoites for the preparation of whole organism vaccines[10]. Such vaccines may provide sterile protection, in spite of the short time sporozoites are extracellular[11]. Notwithstanding the continuous interest in sporozoites as targets for malaria protection, the susceptibility of this infectious stage to chemical intervention is largely undocumented. Current malaria prevention is based on combinations of causal and suppressive prophylactic drugs such as atovaquone–proguanil or sulfadoxine–pyrimethamine[12]. These drugs have been developed several decades ago based on classical pharmacology experiments. Their mode of action in the prevention of human malaria infection has been inferred from rodent models that use *Plasmodium* species that are quite divergent from the species that infect primates[8,13,14]. More recently, humanized mouse models have been introduced that support *P. falciparum* liver- and blood-stage development and pharmacological evaluation of prophylactic compounds[15,16]. By variation of timing of the dosing window, a rough indication can be obtained on whether compounds act at an early or later stage of infection. For example, primaquine appeared to be more efficacious when given 1 day after infection in comparison to administration 4 days after infection[15]. However, the precise developmental processes targeted by causal prophylactics have not been delineated for compounds tested so far.

Robust in vitro systems with sufficient throughput to enable high-throughput screens and extensive dose-response testing against sporozoites have recently emerged. Sporozoites display gliding motility and a capacity to traverse through host cells before reaching their final hepatocyte destination. The gliding of *P. falciparum* sporozoites can be visualized by staining of trails of CSP protein, that is shed on glass slides during movement[17,18]. Traversal of hepatoma cells by *P. falciparum* sporozoites has been visualized by the uptake of a fluorescent dextran dye by wounded cells[19,20]. We have further miniaturized and automated these assay principles using high-content microscopy for quantification. In addition, we have established a transgenic reporter parasite that expresses a GFP-luciferase fusion protein in the sporozoite stages and used this strain to screen chemical libraries

for compounds that affect sporozoite viability. Lastly, we have evaluated compound activity against the process of schizogony in human primary hepatocytes. These combined data provide insight into the stage-specific action of marketed and experimental antimalarials and the drugability of the initial stages of malaria infection.

## Results

**A transgenic reporter strain to monitor sporozoite viability**. We used genetic engineering to insert a GFP-luciferase fusion gene under the control of the CSP promoter in the Pf47 locus of *P. falciparum* strain NF54 (Fig. 1a). FLP-FRT recombination was used to successfully remove the drug selection cassette from the original transfectant (Supplementary Fig. 1a–c), yielding transgenic parasite line NF54-CGL. Following generation of gametocytes that were fed to *Anopheles stephensi* mosquitoes, expression of the chimeric GFP-luciferase reporter was detected by fluorescence microscopy in mature oocysts at day 10 postinfection, and in salivary gland sporozoites isolated from infected mosquitoes at day 17 postinfection (Fig. 1b). In line with these observations, luciferase activity was detected in the whole mosquito homogenates as early as 8 days post feeding, corresponding to late oocyst stage, and increased following full sporozoite development, at 12–14 days post feeding (Fig. 1c). Luciferase activity was not detected in asexual blood-stage parasites, but was very bright in isolated salivary gland sporozoites, as even 10 sporozoites were sufficient to obtain a signal above background (Fig. 1d). The luciferase activity persisted during a 24-h incubation of isolated sporozoites, but decreased when sporozoites were exposed to 1 μM of gramicidin, an ionophore that was previously reported to affect sporozoite viability[21] (Fig. 1e). The $Z'$ value, an indicator for assay robustness[22], was above 0.5 following 24-h incubation, indicating that the assay meets the quality demands of single-point high-throughput screening (Fig. 1e). Variation of sporozoite numbers in the assay showed robust performance with 300 sporozoites per well (Supplementary Fig. 2).

**Miniaturized formats for sporozoite gliding, traversal, and intrahepatocytic schizogony assays**. Sporozoite motility can be visualized by immunostaining of CSP protein that is shed from the surface during the gliding process and leaves a gliding trail. Quantification is often done manually by counting the number of gliding trails from a certain number of microscopic fields on the glass coverslips[17]. For a more efficient assay, we miniaturized it to a 96-well plate format and made it quantitative using an image analysis automated software (Fig. 2a) whereby the signal from immobile sporozoites fixed on the plate is subtracted from the final quantification to only leave the signal due to gliding trails. Gliding motility was sensitive to gramicidin which led to an average 5.7-fold reduction (range 4.0–7.4) in fluorescence signal resulting in an average $Z'$ of 0.44 (range 0.19–0.64) from a total of six independent experiments with 10,000 sporozoites per well (Fig. 2b).

To quantify sporozoite traversal through HC-04 hepatoma cells, we adapted a previously described fluorescent assay format that used Rhodamine Green dextran uptake to visualize cells that were wounded by traversing sporozoites[20]. We used a red fluorescent dye to achieve better distinction from cell autofluorescence and used high-content imaging to quantify the number of hosts cells and the percentage of traversed cells in a 384-well microtiter plate format (Fig. 2c). With 7500 sporozoites per well, signal over background (0.1% DMSO versus 10 μM cytochalasin D) averaged at 20.1 (range 9.3–28.9) with $Z'$ consistently above 0.5 (average 0.68, range 0.59–0.78) in five independent experiments (Fig. 2d).

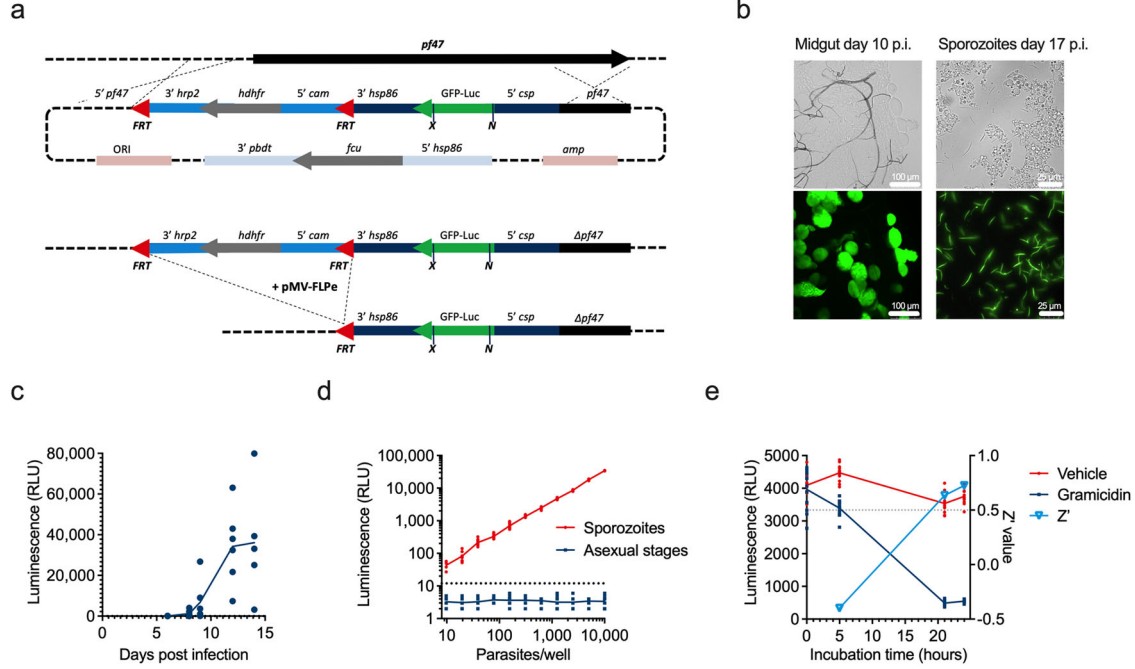

**Fig. 1 A reporter parasite strain to assess sporozoite viability. a** Schematic of the gene targeting strategy to insert a GFP-Luc reporter gene under the control of the *csp* promoter in the pfs47 locus of *P. falciparum* NF54 parasites. The expression cassette was integrated through homologous recombination. Subsequently, the hdhfr selection cassette was removed through FLP-mediated recombination. X (XcmI), N (NcoI): restriction sites; pf47: pf47 locus; hrp: histidine rich protein; cam: calmodulin; hsp: heat shock protein; hdhfr: human dihydrofolate reductase coding region; GFP-Luc: GFP-luciferase fusion protein; fcu: cytosine deaminase/uracil phosphoribosyl transferase; pbdt: *P. berghei* dhfr transcription termination region. **b** Microscope images in phase contrast (upper row) and fluorescence (lower row) channels of mosquito midguts at day 10 postinfection (left panels) and isolated sporozoites at day 17 postinfection (right panels) mounted in PBS on a glass slide. **c** Luciferase activity in infected mosquitoes in time. Mosquitoes were collected 6, 8, 9, 12, and 14 days after infection with a blood meal containing NF54-CGL gametocytes. Luciferase activity was determined in whole mosquito homogenates. Symbols indicate luminescence in individual mosquitoes with 5–6 mosquitoes per timepoint. **d** Luminescence signal of a serial dilution of asexual blood-stage parasites (blue squares) and sporozoites (red dots) of the NF54-CGL parasite strain. The dashed line represents the detection limit in the luminescence assay. Symbols indicate values from 4 to 12 replicates per dilution. **e** Luminescence activity and assay Z′ as a function of time. Isolated salivary gland sporozoites were incubated with vehicle control (0.1% DMSO, red dots) or 1 μM gramicidin (blue squares), and luminescence activity was measured in time. The graph shows data from 12 replicates. These were also used to calculate Z′ values (light blue open triangles). The dotted line indicates Z′ = 0.5.

The development of sporozoites into intrahepatocytic schizonts was monitored by overlaying primary human hepatocytes in 96-well plates with salivary gland sporozoites. After 4 days, parasite schizonts appeared that could be readily identified as Hsp70-positive forms by immunostaining whereas host cells were identified as Hoechst-positive, Hsp70-negative cells (Fig. 2e). High-content imaging was used to enumerate numbers and sizes of parasites and host cells. With 50,000 sporozoites of *P. falciparum* strain NF54 per well, 100 nM atovaquone led to an average 28-fold (range 5.4–73.4) decrease in parasite numbers compared to vehicle control wells, with an average Z′ of 0.41 (range 0.18–0.61) in six independent experiments (Fig. 2f). Given the relatively low infection rate of NF54 parasites, we evaluated the recently described NF175 strain that is, like NF54, of West-African origin and shows drug susceptibility identical to NF54[23,24]. In agreement with published data, sporozoites from strain NF175 gave on average higher infection rates than those of strain NF54 (Fig. 2g), depending on the human donor cells that were used (Fig. 2h)[25]. Development of hepatic schizonts was sensitive to primaquine in all cells tested, suggesting these metabolized the compound into the active form (Fig. 2i). All cells expressed active CYP2D6, the main enzyme responsible for primaquine metabolism, but the activity level did not correlate with primaquine sensitivity (Supplementary Fig. 3). The higher infection rates observed with strain NF175 allowed further miniaturization of the assay to 384-well plates, reducing sporozoite requirements to 10,000 per well (Supplementary Fig. 4a–c). The use of automated dispensing for cell plating reduced

variation between wells (Supplementary Fig. 4d, e). Resulting Z′ values, comparing noninfected to infected wells, averaged at 0.36 (Supplementary Fig. 4f). All three assays relating to sporozoite functions gliding, traversal, and schizont formation were therefore successfully developed as miniature formats allowing a better throughput for compound testing.

**The activity of experimental and marketed antimalarials.** Using the assay panel described above, we explored the activity of a set of 78 marketed and experimental antimalarials. Compounds were tested in duplicate at 10 μM in the sporozoite viability, gliding motility, and traversal assays and at 5 μM for their ability to block intrahepatocytic schizont formation. A small subset showed activity against sporozoite functions prior to hepatocyte invasion (Fig. 3). This subset consisted of a number of 4-aminoquinolines that showed activity across all sporozoite functional assays as well as BIX-01294, a compound targeting the histone methyltransferase. When tested on their effect on liver-stage development at 5 μM, 27 out of 78 compounds blocked schizont formation in human hepatocytes by 90% or more (Fig. 3). We performed more detailed dose-response analyses for these compounds, as well as for those that showed >90% inhibition in one or more of the other assays (hydroxychloroquine, naphtoquine, BIX-01294, and methylene blue). The results revealed potent activity (IC50 < 100 nM) in blockage of intrahepatocytic schizont formation for antifolates P218, cycloguanil, and pyrimethamine, PI4K inhibitors UCT943, UCT944, and UCT048, EF2 inhibitor

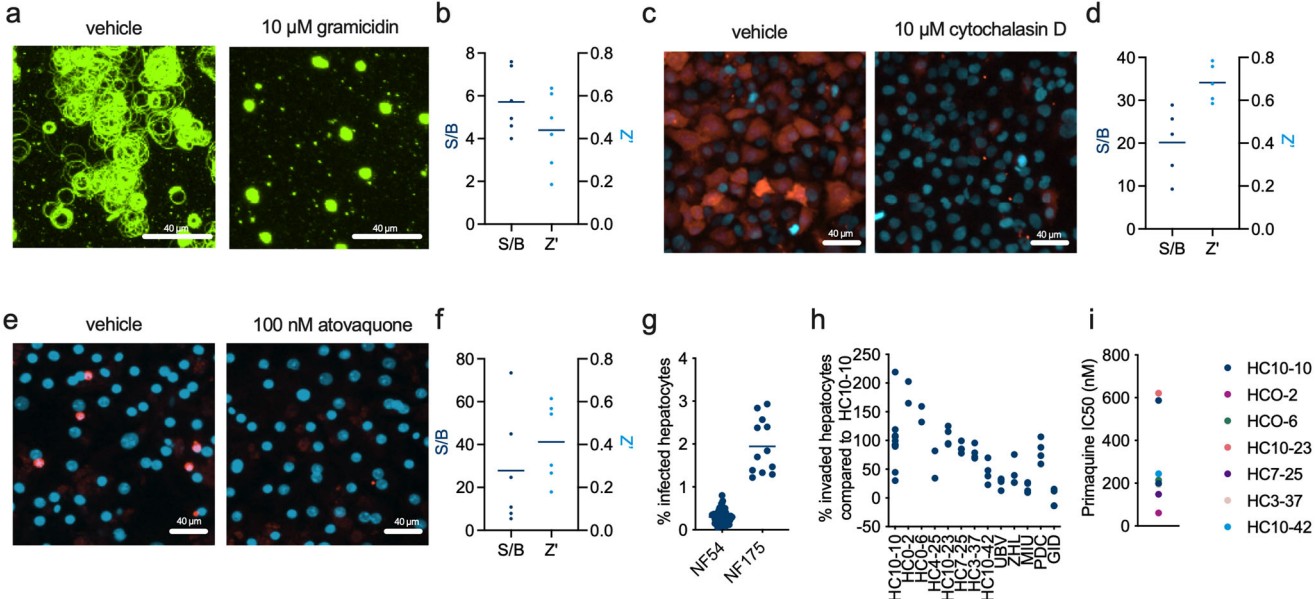

**Fig. 2 Miniaturized assay formats for sporozoite function. a** Micrographs of *P. falciparum* NF54 sporozoite gliding assays showing the difference in gliding activity between sporozoites treated with vehicle (0.1% DMSO) or 10 μM gramicidin. Gliding activity was visualized by immunofluorescent staining of deposited circumsporozoite protein. **b** Signal to background (S/B) and Z′ numbers as determined from independent gliding assays using high-content imaging and quantification of total fluorescence. Each symbol represents an independent experiment, with 2–4 wells for each control (vehicle and gramicidin) **c** Micrographs of *P. falciparum* NF54 sporozoite traversal assays showing the difference in traversal activity between sporozoites treated with vehicle (0.1% DMSO) or 10 μM cytochalasin D. Traversed HC-04 cells were visualized through the addition of Rhodamine Green dextran (red color). Nuclei were stained with Hoechst 33342 (blue color). **d** Signal to background (S/B) and Z′ numbers as determined from independent traversal assays. Each symbol represents an independent experiment, with 2–4 wells for each control (vehicle and cytochalasin. **e** Micrographs of hepatic schizogony assays showing the difference in intrahepatic development between wells treated with vehicle (0.1% DMSO) or 100 nM atovaquone. *Plasmodium falciparum* NF54 parasites were visualized by immunostaining against the hsp70 protein (red color). Nuclei were stained with Hoechst 33342 (blue color). **f** Signal to background (S/B) and Z′ numbers as determined from independent hepatic schizogony assays. Each symbol represents an independent experiment, with 2–4 wells for each control (vehicle and atovaquone). **g** The difference in infection rate between *P. falciparum* NF54 and NF175 strains. The graph shows the percentage of infected cells as determined by quantification of hsp70/Hoechst double-positive forms. Each symbol indicates the average from independent experiments with 2–4 replicates per experiment. **h** Different permissiveness for *Plasmodium* infection of different lots of human primary hepatocytes. Data are from experiments with strain NF175 in 384-well plates and were normalized to the number of infected cells observed in reference lot HC10-10. Symbols indicate individual replicates. **i** IC50 values of primaquine on intrahepatocytic parasite development in primary hepatocytes from different human donors based on 9-point dose-response curves in duplicate. For panels (**a**), (**c**), and (**e**), the white scale bar indicates 40 μm.

DDD498 as well as for KAF-156, atovaquone, and AN13762 (Table 1). In this assay, 8-aminoquinolines primaquine, tafenoquine, and sitamaquine showed slightly weaker activities with IC50s of 235, 930, and 412 nM, respectively. The 4-aminoquinolines showed poor activity against intrahepatocytic schizont formation with IC50s of 869 nM for pyronaridine and in the micromolar range for amodiaquine and the N-desethyl form of this compound. None of the compounds in the entire tested set of marketed and experimental compounds showed potent activity against sporozoite viability, gliding motility, or traversal. Pyronaridine, N-desethyl amodiaquine, and BIX-01294 showed weak activity in these assays. However, IC50 values were considerably higher than those observed against asexual blood-stage parasites, making it unlikely that doses used for malaria treatment and their cognate levels of compound circulating in the blood could contribute to a prophylactic activity against sporozoites. Our data show that compounds that are currently used as causal prophylactics (e.g., atovaquone, pyrimethamine, primaquine) or in clinical development for this purpose (e.g., KAF-156, P218) act at the level of hepatic schizogony and not earlier in the infection. We further exemplified this with a detailed study of P218, a candidate for malaria chemoprotection[26]. The compound was inactive when administered in the first 3 h of the invasion and development process that leads to hepatic schizonts (Fig. 4). However, it did suppress schizont formation when administered

from 3 to 48 h postinfection or at a later stage (48–96 h postinfection) of schizont formation. Relative to the vehicle control, the reduction in parasite numbers averaged 98% when the compound was present during the entire course of the experiments and was slightly lower (86–90%) for shorter incubations. Parasites that did develop under drug pressure were smaller in size. It appeared that exposure of the parasites to P218 in the first 48 h showed slightly larger remaining parasites in comparison to exposures during the last 48 h. This may suggest that P218 effects are reversible and that parasites resume growth when compound levels drop.

**High-throughput screening of two libraries for sporontocidal activity**. In order to identify new chemotypes with action against sporozoites, we carried out a high-throughput screening campaign of two chemical libraries, namely the Global Health Chemical Diversity Library (GHCDL) and the reFRAME library[27]. A total of ~81,000 compounds was screened in the sporozoite viability assay in a 384-well format with 1000 sporozoites per well. For both libraries, we followed a process of initial screening at a single concentration in a single well (10 μM for GHCDL and 2.5 μM for reFRAME), followed by a confirmation screen at the same concentration, either in a single well for GHCDL or in triplicate for reFRAME (Fig. 5a). Approximately 50% of

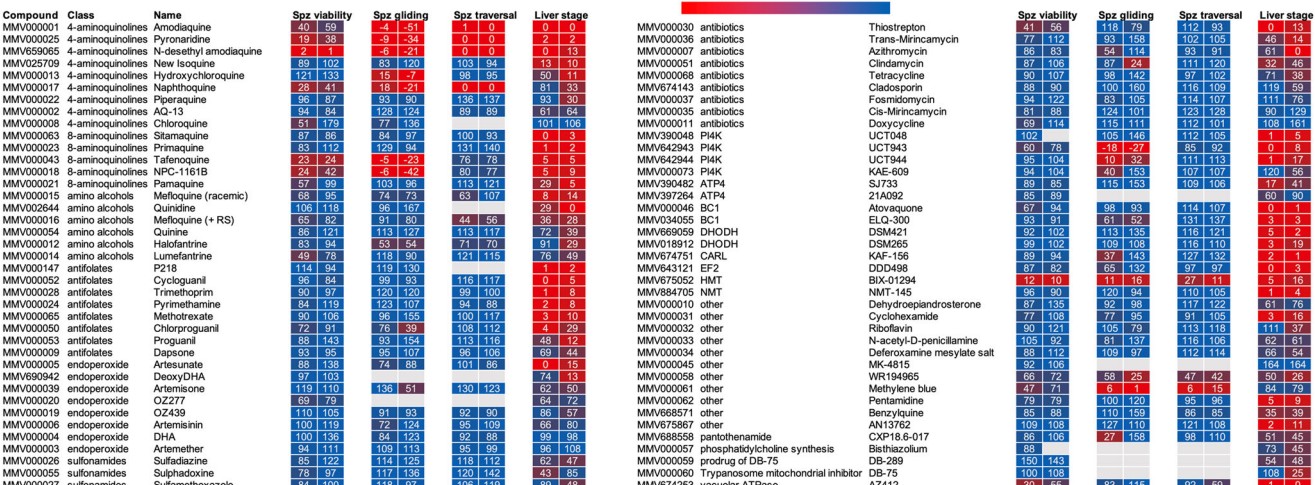

**Fig. 3 Activity of marketed and experimental antimalarials in sporozoite viability, gliding, traversal, and hepatic schizogony assays.** Compounds were tested against *P. falciparum* NF54 parasites using hepatocyte donor HC10-10 in the liver-stage assays. The figure shows percentage viability, gliding motility, cell traversal capacity, and a number of liver-stage parasites relative to the vehicle controls (0.1% DMSO). Compounds were tested at 10 μM except for the liver-stage assay, where compounds were assayed at 5 μM. The figure shows average data from two independent experiments with two replicates per experiment. Spz: sporozoite, DHA: dihydroartemisinin, PI4K: phosphatidylinositol 4-kinase, ATP4: P-type Na+ adenosine 5'-triphosphatase 4, BC1: coenzyme Q: cytochrome c–oxidoreductase, DHODH: dihydroorotate dehydrogenase, CARL: cyclic amine resistance locus, EF2: elongation factor 2, HMT: histone methyl transferase, NMT: N-myristoyl transferase.

**Table 1 IC50 values of selected compounds in *P. falciparum* NF54 sporozoite functional assays.**

| | | | IC50 values (nM) | | | | |
|---|---|---|---|---|---|---|---|
| MMV code | Class | Name | ABS | Spz viability | Spz gliding | Spz traversal | Liver stage |
| MMV659065 | 4-aminoquinolines | N-desethyl amodiaquine | 12 | 1635 | 1510 | 472 | 1146 |
| MMV675052 | HMT | BIX-01294 | 75 | 3772 | >10,000 | 4021 | <5000 |
| MMV000017 | 4-aminoquinolines | Naphthoquine | 3.2 | 4363 | 2921 | 1094 | >5000 |
| MMV000001 | 4-aminoquinolines | Amodiaquine | 3.9 | 6311 | 2325 | 230 | 3759 |
| MMV000025 | 4-aminoquinolines | Pyronaridine | 4.9 | 6471 | 836 | 458 | 869 |
| MMV000018 | 8-aminoquinolines | NPC-1161B | 419 | 7569 | 9808 | >10,000 | 1454 |
| MMV000043 | 8-aminoquinolines | Tafenoquine | 631 | 8614 | >10,000 | >10,000 | 930 |
| MMV000013 | 4-aminoquinolines | Hydroxychloroquine | 16 | >10,000 | >10,000 | >10,000 | >5000 |
| MMV000023 | 8-aminoquinolines | Primaquine | 1191 | >10,000 | >10,000 | >10,000 | 235.0 |
| MMV000063 | 8-aminoquinolines | Sitamaquine | | >10,000 | >10,000 | >10,000 | 412.0 |
| MMV000030 | Antibiotics | Thiostrepton | 388 | >10,000 | >10,000 | >10,000 | 1349 |
| MMV000147 | Antifolates | P218 | | >10,000 | >10,000 | >10,000 | 0.3 |
| MMV000052 | Antifolates | Cycloguanil | 4.5 | >10,000 | >10,000 | >10,000 | 3.4 |
| MMV000024 | Antifolates | Pyrimethamine | 17 | >10,000 | >10,000 | >10,000 | 37.0 |
| MMV000028 | Antifolates | Trimethoprim | 940 | >10,000 | >10,000 | >10,000 | 292.0 |
| MMV000065 | Antifolates | Methotrexate | | >10,000 | >10,000 | >10,000 | 569.2 |
| MMV000046 | BC1 | Atovaquone | 0.6 | >10,000 | >10,000 | >10,000 | 8.3 |
| MMV034055 | BC1 | ELQ-300 | 4.8 | >10,000 | >10,000 | >10,000 | 110.2 |
| MMV674751 | CARL | KAF-156 | | >10,000 | >10,000 | >10,000 | 16.2 |
| MMV669059 | DHODH | DSM421 | | >10,000 | >10,000 | >10,000 | 505.0 |
| MMV643121 | EF2 | DDD498 | 1 | >10,000 | >10,000 | >10,000 | 1.1 |
| MMV000005 | Endoperoxide | Artesunate | 3.5 | >10,000 | >10,000 | >10,000 | >10,000 |
| MMV884705 | NMT | NMT-145 | | >10,000 | >10,000 | >10,000 | 430.0 |
| MMV675867 | Other | AN13762 | | >10,000 | >10,000 | >10,000 | 5.9 |
| MMV000031 | Other | Cyclohexamide | | >10,000 | >10,000 | >10,000 | 843.8 |
| MMV000062 | Other | Pentamidine | | >10,000 | >10,000 | >10,000 | 974.9 |
| MMV000061 | Other | Methylene blue | NA | >10,000 | 3279 | 3522 | >5000 |
| MMV642944 | PI4K | UCT944 | | >10,000 | >10,000 | >10,000 | 1.4 |
| MMV642943 | PI4K | UCT943 | NA | >10,000 | >10,000 | >10,000 | 4 |
| MMV390048 | PI4K | UCT048 | 28 | >10,000 | >10,000 | >10,000 | 45.1 |
| MMV674253 | Vacuolar ATPase | AZ412 | 12 | >10,000 | >10,000 | >10,000 | 3040 |

IC50s were calculated from 9-point dose-response curves with two replicates per data point. Asexual blood stage (ABS) data was taken from a previous publication[13].

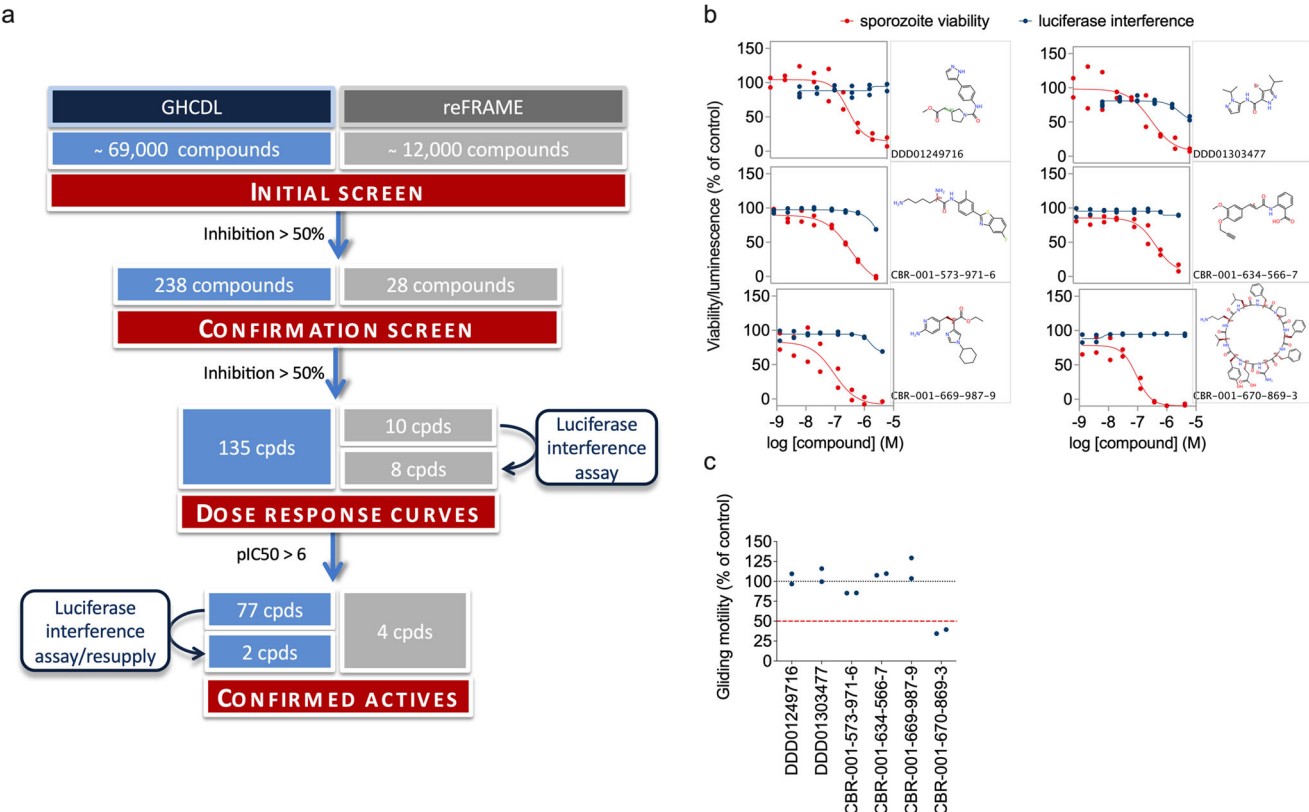

| | Exp. 1 | | | Exp. 2 | | |
|---|---|---|---|---|---|---|
| | IC50 (nM) | max reduction parasite number (%) | relative size remaining parasites (%) | IC50 (nM) | max reduction parasite number (%) | relative size remaining parasites (%) |
| | >10,000 | 0 | 87 | >10,000 | 0 | 100 |
| | 2.9 | 86 | 50 | 3.3 | 86 | 54 |
| | 2.9 | 97 | 14 | 2.8 | 99 | 19 |
| | 2.9 | 86 | 50 | 3.2 | 87 | 45 |
| | 2.8 | 97 | 34 | 3.1 | 98 | 16 |
| | 5.2 | 90 | 30 | 5.2 | 88 | 28 |

**Fig. 4 Mode of action of P218 on sporozoite hepatocyte invasion and schizont maturation.** *Plasmodium falciparum* NF175 sporozoites/infected HC10-10 hepatocytes were incubated with serial dilutions according to the scheme indicated on the left of the figure. Blue bars indicate the presence of the compound, with daily refreshments for longer incubations. The table indicates IC50 values determined in two independent experiments, with two replicates per experiment. In addition, the table lists the maximum reduction in parasite numbers that were achieved (plateau of the dose-response curve), and the relative sizes of remaining parasites in comparison with vehicle controls (0.1% DMSO).

**Fig. 5 High-throughput screening in sporozoite viability assays. a** Schematic representation of the high-throughput screenings carried out with the GHCDL and the reFRAME library with our luminescence-based sporozoite viability assay and luciferase interference assay. **b** Viability of sporozoites after 24-h incubation (red dots) and recombinant luciferase signal after a 1-h incubation (blue dots) with different doses of the hits from the GHCDL and reFRAME library. Activity is expressed as the percentage of the positive and negative control luminescence signal. The figure shows data from two replicate experiments. **c** Quantification of the gliding trails of sporozoites treated with a single 10 µM and 3.3 µM dose of hit compounds from the GHCDL (DDD compound codes) and reFRAME (CBR compounds) library, respectively. Error bars indicate standard deviations from two replicates, which are shown by individual symbols.

compounds with initial activity was confirmed at this stage and progressed toward dose-response analyses. In addition, false positives were eliminated by testing for direct inhibition of luciferase activity using a recombinant form of the enzyme. For the reFRAME library, eight compounds out of the ten tested did not interfere with luciferase. Of these, four compounds showed a pIC50 > 6 in the sporozoite viability dose-response analysis. For the GHCDL, of the 77 compounds selected in the dose-response curve steps, five were out of stock, 12 did not confirm in the sporozoite viability assay after the resupply of the compound

(i.e., IC$_{50}$ > 1 µM) and 58 interfered with luciferase. At the end of the screens, the different steps led to the identification of two and four confirmed actives from the 69,000 and 11,000 compounds of the GHCDL and reFRAME libraries, respectively (Fig. 5a). Although none of these compounds showed inhibition of recombinant luciferase greater than 50% at the highest concentration tested, some showed weak inhibition of the enzyme, casting doubt on the specificity of the result from the sporozoite viability assay (Fig. 5b). Moreover, five of the confirmed actives did not inhibit sporozoite gliding motility at 10 µM (Fig. 5c),

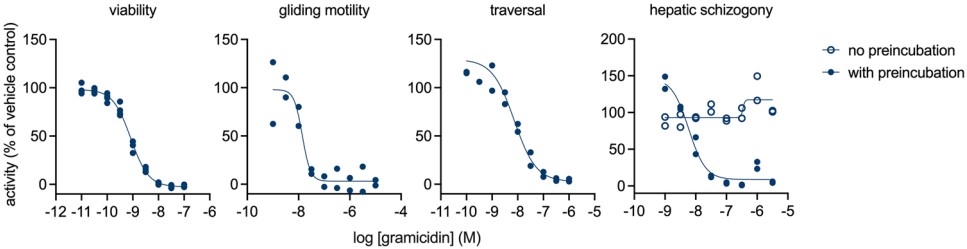

**Fig. 6 Sporontocidal activity of gramicidin.** The figure shows dose-response analyses of gramicidin in sporozoite viability, gliding, traversal, and hepatic schizogony assays. In the latter assay, sporozoites were pre-incubated with gramicidin for 30 min prior to hepatocyte infection, or directly added to the host cells. The figures show data normalized to the respective positive and negative controls in the assays. Each condition was tested in 2–4 replicates.

indicating that the effect observed in the sporozoite viability assay did not translate into a functional defect in motility. This may be explained by a difference in assay temperature and in the length of treatment between the two assays. However, a drug acting on the sporozoite stage should be fast-acting as sporozoites only stay a couple of hours in the body before changing into the liver stage. The gliding assay may therefore be a better indicator of the compound potential as future prophylaxis drugs, and only compounds that interfered with gliding were considered for further characterization. A single compound, CBR-001-670-869-3, derived from the reFRAME repurposing library, met this criterion as it reduced sporozoite gliding motility (Fig. 5c). This compound was annotated as tyrothricin, a mixture of cyclic decapeptides including gramicidin[28], that was also used as a control in this assay. Even though the hit rate was very low in this screening, this last finding validates the screening cascade as the one compound related to our control could successfully be identified.

**Rapid killing of sporozoites prevents liver schizogony.** It has been previously reported that gramicidin is the main active component of tyrothricin that inhibits the development of liver-stage parasites[21,29], but it is not clear whether the compound acts at the level of schizogony or against nonreplicating parasites preceding this process. To investigate this in more detail, we performed a full dose-response analysis of gramicidin in the sporozoite functional assays, and in the intrahepatocytic schizogony assay. The results show that gramicidin affects sporozoite viability, gliding motility, and traversal capacity with IC50 values of 0.8, 14, and 8 nM, respectively (Fig. 6). It prevented schizogony in primary hepatocytes only when the compound was pre-incubated with sporozoites prior to the invasion and the compound was inactive when it was added 3 h after sporozoites were overlaid on the hepatocytes. The combined data indicate that gramicidin is the sole compound identified in this study that prevented hepatic schizogony by directly acting on the sporozoite.

## Discussion
Sporozoites may be attractive drug targets as they are the primary source of infection in humans, form a population bottleneck in the life cycle of malaria parasites, and are the longest free-living form of the parasite[21]. Using a genetically engineered strain of *P. falciparum* parasites that expresses a GFP-luciferase fusion protein under the CSP promoter, we developed a high-throughput screening assay for malaria sporozoite viability. In addition, we used miniaturized assay formats to study sporozoite gliding motility, cell traversal, and hepatic schizogony. Even though we used miniaturized formats for the latter assays, they require relatively high sporozoite numbers per well. In our lab, experimental infections yield ~40,000 salivary gland sporozoites per mosquito[30,31]. Screening of the 81,000 compounds from the ReFrame and GHDL libraries in the sporozoite traversal assay

would require the dissection of 15,188 mosquitoes, controls not included. The luminescence-based viability assay as presented herein has numerous advantages: it requires a small number of parasites to give a signal above background and offers a consistent, fast, and direct readout of viability. At 1000 sporozoites per well, screening 8100 compounds requires the dissection of 2025 mosquitoes. Using the viability assay, we screened a set of compounds with antimalarial activity and two chemical libraries. In all compound sets, the number of active compounds was surprisingly low, and in the GHCD library in particular, screening mainly led to the identification of false positives. The difficulty to target sporozoites with existing antimalarial compounds may be in line with a proteomic study that reported 49% of the proteins expressed in sporozoites were specific to this stage[32], making the sporozoite a special target in the malaria cycle. Alternatively, the permeability of the sporozoite membrane may differ from the other life cycle stages of the parasite. In the validation set, 4- and 8-aminoquinolines showed weak activity against sporozoites. Compounds of the 4-aminoquinoline class are thought to act on the asexual blood stage through interference with heme crystal formation in red blood cell hosts[5]. In sporozoites, this mode of action is therefore irrelevant, which may explain the dramatic shift in activity between the two different parasite stages. As for the 8-aminoquinolines, their exact mode of action and active metabolites remain under debate, but they are known to be active on other nondividing stages of malaria in vivo, such as gametocytes and hypnozoites while exhibiting poor activities in vitro systems that lack metabolizing activity[13,33]. Our observation that primaquine inhibited liver schizogony in infected primary hepatocytes suggests that these cells are metabolically active, although we did not observe a correlation between primaquine IC50 and CYP2D6 activity. Given the observed donor-depended variation in primaquine activity, the evaluation of compounds that require metabolic activation may benefit from using pooled human primary hepatocytes. In the absence of the final metabolite in the sporozoite viability, gliding, and traversal assays, they may underestimate the activity of 8-aminoquinolines. Other antimalarial compounds that are known to act on nondividing mature gametocytes in vitro failed to decrease sporozoites viability, outlining again the particularity of sporozoites. This is the case for instance for the class of compounds that targets PI4K, a kinase involved in lipid metabolism, or ATP4, associated with ion homeostasis[33]. Likewise, compound KAF-156, for which the mode of action is unknown but which has multistage activity, including efficacy in a prophylaxis mouse model, failed to reduce sporozoite viability. Compounds such as atovaquone and pyrimethamine, targeting the mitochondrial BC1 complex and dihydrofolate reductase (DHFR), respectively, are established as causal malaria prophylactics. They have been shown to block the development of *P. falciparum* sporozoites into hepatic schizonts but their precise mode of action against human parasites is not documented in detail[29,34]. Our data indicate that they have no

activity against sporozoites but block intrahepatocytic schizogony. Similarly, P218, a DHFR inhibitor that is currently in clinical development as a chemoprotective antimalarial agent[35], did not interfere with sporozoite viability or motility but blocked schizogony in primary human hepatocytes. Our data indicate an IC50 of 0.3 nM against NF54 parasites and around 3 nM against NF175. These IC50 values are in good agreement with previous data showing an IC50 < 12 nM against liver-stage parasites[29]. Our data indicate that exposure for 2 days is sufficient to achieve a reduction of 86–90% in parasite numbers, even in cases when the compound was added to cultures with pre-existing parasites. This suggests that the compound exerts a cidal activity.

Sporozoite drug discovery is in its infancy. It is perhaps not surprising that compounds that were discovered and developed against dividing blood-stage parasites do not act against the non-dividing sporozoites. Similarly, most of the marketed antimalarials exert poor activity against nondividing mature gametocytes[13,36]. For a long time, this stage has been considered quiescent and not druggable[37]. The availability of miniaturized screening assays has changed this image and these days a rich compendium of molecules with gametocytocidal activity is available[33,38]. Likewise, sporozoite drug discovery will benefit from miniaturized assays described here and in previous work. A target-based approach to screen for inhibitors of the interaction between thrombospondin-related anonymous protein and aldolase identified a compound with micromolar activity against P. berghei sporozoite gliding[39]. KNX-115, an inhibitor of Plasmodium falciparum MyoA, blocks sporozoite traversal and hepatocyte invasion[40]. Screening of the MMV pathogen box identified a handful of compounds with sub-micromolar IC50 against sporozoite gliding motility[18]. Another study elegantly demonstrated the suitability of live imaging to quantify the motility of P. berghei sporozoites with as little as 2000 parasites per well in 384-well plates and identified hit compounds with micromolar activity from the MMV malaria box and a set of FDA-approved drugs[21]. Together these observations suggest that the machinery underlying sporozoite motility is a viable target for malaria drug discovery. Further exploration of this theme may benefit from cross-screening compounds targeting motility in other eukaryotic cells, e.g., from programs targeting cancer metastasis[41].

A number of studies have shown that ionophores like monensin and gramicidin exert potent activity against all life cycle stages of malaria parasites[21,29,42,43]. The detailed profiling of gramicidin presented here shows that the compound affects sporozoite viability, motility, and their ability to form hepatic schizonts with nanomolar IC50 values. As such, it is the most potent compound targeting sporozoites described to date. Although monensin, an ionophore related to gramicidin, is used in veterinary applications for the treatment of coccidiosis, its toxicity precludes development for human use[44,45]. They are however proof of concept for this screening method which starts with a high-throughput, luminescence-based assay of sporozoite viability, and is completed with optimized, quantitative assays for sporozoite gliding, traversal, and intrahepatic development. This screening cascade may thus contribute to the identification of novel molecules to fill the need for chemoprotection in the fight against malaria.

## Methods

**Chemical library and compound sources**. The malaria validation set is composed of 78 antimalarial compounds marketed or in a late stage of discovery and was composed by the Medicines for Malaria Venture (MMV).

The Global Health Chemical Diversity Library (GHCDL) is a 69,000-compound library composed of commercially available and lead-like compounds as described elsewhere[46]. The reFRAME library aims at drug repurposing and is composed of ~12,000 compounds that have either reached the market or been studied as leads in other drug screening campaigns[27]. Gramicidin and DMSO were from Sigma-Aldrich.

**Parasites and culture**. Asexual blood stages of the P. falciparum parasite lines were cultured in RPMI 1640 medium supplemented with 367 μM hypoxanthine, 25 mM HEPES, 25 mM sodium bicarbonate, 5% human type O red blood cells and 10% human type A serum in a semi-automated system[47]. For gametocyte production, cultures were inoculated with 1% asexual blood-stage parasites and cultured for 2 weeks with twice daily medium replacement. At 14–16 days postinoculation, cultures were fed to Anopheles stephensi mosquitoes through membrane feeders and mosquitoes were maintained at 27 °C and 80% relative humidity for 2 weeks. To support sporozoite maturation, mosquitoes received an extra blood meal after 1 week. At day 14–21 postinfection, salivary glands were dissected from the mosquitoes in L-15 (Leibovitz) medium (Lonza) and sporozoites were isolated through homogenization in a homemade glass grinder[17].

**Generation of NF54-CGL parasite line**. A 1.2-kb fragment of the csp (circumsporozoite) gene promoter was amplified from P. falciparum NF54 genomic DNA by PCR using primers TGAGGTACCTGTTTGAGCTTATTTCAATTGTTGTG and CTCATCATACCGGTTAATTTATAATATACGTGGTTTC, cloned into a reporter GFP:luciferase reporter plasmid[48] using KpnI/AgeI restriction digestion and ligation and introduced into the pfs47 locus of P. falciparum strain NF54 by electroporation. To this end, ring-stage parasites at an approximate parasitemia of 5% were electroporated with 80 μg plasmid DNA at 0.31 kV, 960 μF in a 0.2 cm cuvette using an Electro Cell Manipulator 600 (BTX, Holliston, USA). Transfected parasites were placed in RPMI 1640 medium supplemented with 367 μM hypoxanthine, 25 mM HEPES, 25 mM sodium bicarbonate, 5% human type O red blood cells, and 10% human type A serum. WR99210 was added to a final concentration of 2.6 nM to select the presence of the Hsdhfr selectable marker. WR99210-resistant transfectants were cloned by limited dilution and transferred to the semi-automated culture system and cultured for further phenotype and genotype analyses. Following confirmation of expression of the reporter fusion protein, the dihydrofolate reductase selection marker was removed through FLP-mediated recombination by transfection with plasmid pMV-FLPe[48] and selection on 5 μg/ml blasticidin to select for the pMV-FLPe plasmid. The integrity of the genomic integration site was checked by PCR analyses and restriction digestion. The resulting reporter strain was named NF54-CGL. Sensitivity to the antifolate drug pyrimethamine was assessed through asexual blood-stage replication assays[48]. Briefly, parasites were seeded at 0.4% in RPMI 1640 medium supplemented with 367 μM hypoxanthine, 25 mM HEPES, 25 mM sodium bicarbonate, 1.5% human type O red blood cells, and 10% human type A serum in 30 μl in a 384-well black view plate (VWR). Then, 30 μl of compound diluted in the medium was added and parasites were incubated for 72 h at 37 °C, 3% $O_2$, 4% $CO_2$, and 93% $N_2$. Subsequently, DNA was stained by adding SYBRGreen reagent diluted 10,000 fold according to the instructions of the manufacturer (Invitrogen). Relative parasitemia was quantified by reading fluorescence on a Biotek Synergy 2 plate reader (Biotek, Winooski, VT).

**GFP and luciferase expression analysis**. In order to monitor the presence of NF54-CGL parasites in infected mosquitoes, GFP expression in the isolated midguts, salivary glands, and sporozoites was visualized and photographed on a fluorescence microscope (Zeiss) with a digital camera. To this end, midguts of infected mosquitoes were isolated and mounted on glass slides in a droplet of PBS 7–10 days postinfection with NF54-CGL. Seventeen days postinfection, the salivary glands of infected mosquitoes were isolated and either mounted on a glass slide or ground in a glass mortar to free the sporozoites. To assess luciferase expression, 45 μl homogenate from whole mosquitoes was mixed with 45 μl of Bright-Glo reagent (Promega, Madison, WI) and luminescence was measured in a Biotek Synergy 2 plate reader.

**Sporozoite viability assay**. Fourteen to twenty-two days postinfection with NF54-CGL, salivary glands of infected mosquitoes were isolated in L-15 (Leibovitz) medium (Lonza) supplemented with penicillin-streptomycin and L-glutamine and ground in a glass mortar to free the sporozoites. After determining the sporozoite count with a hematocytometer, sporozoites were diluted to the desired concentration with L-15 medium supplemented with penicillin-streptomycin and L-glutamine without (incomplete medium) or with (complete medium) 10% heat-inactivated fetal bovine serum (hi-FBS). For the screenings, 12 μl/well of a 25,000–83,000 sporozoite/ml dilution (corresponding to 300–1000 sporozoites/ well) were dispensed using a Multidrop (Thermofisher) or an electronic multichannel (Ovation) in white 384-well plates (Proxiplate 384-TC, Perkin Elmer) prespotted with the chemical library. For assay development, a test of antimalarial drugs, and for compounds issued from resynthesis, 6 μl of a 167,000 sporozoite/ml dilution (corresponding to 1000 sporozoites/well) were dispensed using an electronic multichannel in white 384-well plates containing 6 μl of compounds diluted at twofold the final concentration in incomplete or complete medium. In all cases, the compounds had been dissolved in DMSO, and the final DMSO concentration in the assay was 0.1%. Each plate contained vehicle (0.1% DMSO) and positive (100 nM to 1 μM gramicidin) controls, which were used for the normalization of the signals. The plates were incubated at 26 °C with ambient gas in a humidity-controlled incubator for 24 h, then 6 μl of Bright-Glo reagent (Promega) was added to the wells. After a 3-min incubation, luciferase activity was measured using a Synergy 2 multi-purpose plate reader.

**Luciferase interference assay.** Recombinant luciferase at 10–15 mg/ml (Promega) was diluted $10^6$ times in complete medium and dispensed, 12 µl/well, in white 384-well plates (Proxiplate, Perkin Elmer) pre-spotted with compounds. After 1 h of incubation, 6 µl of Bright-Glo reagent (Promega) was added to the wells. Following a 3-min incubation, luciferase activity was measured using a Synergy 2 multi-purpose plate reader (Biotek, Winooski, VT). Each plate contained vehicle controls (0.1% DMSO) and wells that contain either no luciferase or no Bright-Glo. The average of these controls was used for the normalization of the compound signals.

**Quantitative sporozoite motility assay.** A 96-well plate with a glass bottom was coated with 3SP2 antibody[19] at a final concentration of 5 µg/ml in PBS. Sporozoites were isolated from infected mosquitoes and diluted to a final concentration of 400,000 sporozoites/ml in L-15 (Leibovitz) medium. Then, 30 µl of this dilution was dispensed in a V-shaped 96-well plate containing 30 µl of compound dilution in L-15 medium with 20% fetal calf serum (10% final concentration). This plate was pre-incubated for 30 min at room temperature (20 °C). Meanwhile, the coated glass-bottom plate was washed with PBS and blocked with L-15 medium with 10% fetal calf serum. Following the incubation, the medium was removed and replaced with 50 µl of the compound-sporozoite mixture. The plate was centrifuged at 3000 rpm for 5 min, before starting incubation for 90 min at 37 °C with 4% $CO_2$ and 3% $O_2$. The plate was then washed with PBS and fixed with paraformaldehyde. The gliding trails were stained with 1 µg/ml biotinylated 3SP2 antibody and subsequently with 10 µg/ml streptavidin conjugated with an Alexa Fluor 594 dye (Life Technologies). The fluorescent signal was visualized and automatically photographed using a Cytation plate reader (Biotek). The resulting images (16 images per well stitched together) were analyzed using a macro running on Image J. First, a cutoff value is defined by the user for each batch of pictures to delineate the overexposed areas corresponding to sporozoites left in the wells. Second, the command autothreshold was run on the original image with default parameters to outline all signals above the background. The final processed image, from which the quantification can be done, results from the subtraction of the cutoff image from the thresholded image to leave only the gliding trails.

In gliding motility experiments with hit compounds from the high-throughput screening campaign, a prolonged pre-incubation time of 4 h was used in order to identify potential slow-acting compounds. Here, the temperature during the pre-incubation was performed on ice in order to preserve sporozoite motility during the pre-incubation.

**Traversal assay.** Human HC-04 hepatoma cells were seeded in DMEM/F12 medium with 10% heat-inactivated fetal bovine serum and 100 U/ml penicillin and 100 µg/ml streptomycin in 384-well clear bottom black plates and grown to 100% confluency over the course of 2 days at 37 °C and 5% $CO_2$. Diluted compounds were pre-incubated with salivary gland sporozoites (7500 per well) for 30 min at room temperature (20 °C) in 55 µl HC-04 culture medium. Subsequently, 5 µl rhodamine-dextran was added to a final concentration of 0.5 mg/ml. The medium was removed from the HC-04 cells and they were overlaid with 50 µl of the sporozoite-compound-dextran mixture. The plates were centrifuged at 1900 × g for 10 min and incubated for 1 h at 37 °C and 5% $CO_2$. Subsequently, the cells were washed and fixed with 4% paraformaldehyde for 15 min at room temperature (rT). Hereafter, cells were counterstained for 15 min at rT with DAPI (4',6-Diamidino-2-Phenylindole, Dihydrochloride) in PBS. Cells were washed and the total number of cells and the number of traversed cells were quantified using an ImageXpress Pico automated cell imaging system (Molecular Devices).

**Liver-stage development.** Intrahepatocytic development of *P. falciparum* NF54 parasites was studied by seeding primary human hepatocytes at 60,000 cells/well and culturing for 2 days in collagen-coated 96-well plates. They were then overlaid with 50,000 *P. falciparum* NF54 sporozoites and compounds. The supernatant was refreshed daily with fresh compounds. Four days postinfection, parasites were visualized by fluorescent (Alexa Fluor 546, Invitrogen) immunostaining of the Hsp70 antigen, and host and parasite nuclei were visualized by Hoechst 33342 DNA staining. Parasites and host cells were quantified and sized using an ImageXpress Pico high-content microscope using Cell Report Xpress software. Object parameters for quantification of intensity and area were calculated automatically based on user-selected cells of interest and were adapted, where necessary, to achieve a clear distinction between positive (100 nM atovaquone) and negative (0.1% DMSO) control wells. Recognition of parasites was based on the overlap of Hoechst and Alexa Fluor 546 signals. Using parasite strain NF175[25], the assay was miniaturized to a 384-well plate format. Procedures were similar except that 18,000 primary hepatocytes were seeded per well and, following culturing for 2 days, overlaid with 10,000 sporozoites.

**Statistics and reproducibility.** $IC_{50}$ values were calculated by applying a four-parameter logistic regression model using a least-squares method to find the best fit, using the Graphpad Prism 5.0 software package. To assess assay performance,

$Z'$ values[22] were calculated using the following formula:

$$Z' = 1 - \frac{3\sigma_{sample} + 3\sigma_{control}}{mean\ of\ sample - mean\ of\ control} \qquad (1)$$

Reproducibility was assessed by measuring independent experimental replicates. The number of replicates ranged from two per condition for regression analyses involving multiple conditions, e.g., dose-response analyses, to 12 replicates for single-point analyses, e.g., $Z'$ calculations.

**Reporting summary.** Further information on research design is available in the Nature Portfolio Reporting Summary linked to this article.

## Data availability
Data used to make figures are available in Supplementary Data 1.

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

## Acknowledgements

Louis Wolf is kindly acknowledged for FIJI script programming. The authors thank Laura Pelser-Posthumus for help with mosquito husbandry and sporozoite production.

## Author contributions

M.M. generated and analyzed data and drafted and edited the manuscript. J.M.B., M.W.V., K.J.K.M., M.d.B., and D.S.R. generated and analyzed data. E.C., M.K., and D.G. provided and managed compound libraries. B.C., R.W.S., and K.J.D. conceived the studies, supervised experiments, and edited the manuscript.

## Competing interests

R.W.S. and K.J.D. hold stock in TropIQ Health Sciences B.V. All other authors declare no competing interests.
