## [Peer Review File · Communications Biology]

Reviewers' comments:

Reviewer #1 (Remarks to the Author):

The manuscript by Miglianico, Bolscher and colleagues describes the development of a high-throughput screening pipeline to test the feasibility of targeting the sporozoite stage of *P. falciparum* through chemical interventions. This is an interesting concept that has only recently started to gain some traction, as nicely outlined in the Discussion. The authors established their screening pipeline by adapting well known assays, each measuring distinct sporozoite properties (gliding motility, cell traversal, hepatocyte invasion/development into exoerythrocytic forms), for high-content imaging, feature extraction and quantification. The manuscript also details the development of a new bioluminescence-based assay to measure sporozoite viability. As proof-of-concept, the authors screened two large compound libraries using the viability assay and an additional 78 compounds with known antimalarial activity using the entire pipeline. The output of these screening campaigns was somewhat underwhelming, as only a few hits show activity against the sporozoite stage. It is nevertheless reassuring that the single hit in the viability assay has been identified as an inhibitor of sporozoite motility in a previous study (reference 25). The authors conclude that the liver stage is more easily druggable than the sporozoite stage, which might indeed be the case and for several reasons, including, as they put forward, the unique proteome of this parasite stage. Is it possible that another reason might lie with differences in drug permeability through the membrane? Given the poor hit rates reported here, would future screens gain from using more focused compound libraries that tap into sporozoite-enriched biological pathways, such as cell motility?

Overall, this is a well written manuscript that describes a screening approach designed to enable the identification of compounds offering chemoprotection against malaria. The authors should discuss this concept and provide references for chemoprevention strategies already being implemented in the field.

Specific comments:

1. Fig. 2 A shows the result of a gliding assay in which sporozoites were subject to a 4-hour pre-incubation at 4°C before the actual assay at 37°C. It is not clear why this pre-incubation step was necessary. The cell traversal assay, which has a similar duration (60 vs 90 min), was carried out without this prolonged pre-incubation. Could the authors provide a rationale for why these two assays use a different drug incubation protocol?
2. Do the authors observe any changes in sporozoite viability after the pre-incubation step at 4°C? Does it adversely affect sporozoite motility or their ability to infect hepatocytes?
3. The switch to NF175 parasites in Fig. 2 G-I is somewhat confusing. Perhaps those panels could be moved to a Supplementary Figure, as they mainly reinforce previous reports that donor-dependent variation in infection is to be expected (<https://doi.org/10.15252/embj.2020106583>).
4. The authors show in Fig. 2 H-I that the hepatocyte permissiveness to infection is donor dependent and that the IC50 of primaquine also varies from donor to donor. In the screen of experimental and marketed antimalarials shown in Fig. 3, it is not clear if the hepatocytes used in each replicate originated from the same or different donors (please specify this point in the legend of both Fig. 3 and Fig. 4). Do the authors foresee the need to implement steps, such as pooling different hepatocyte donors or validate hits across donors, to account for this natural variability?
5. The hit rate in the viability assay was extremely low, and while the authors offer several good explanations for this outcome, is it possible that the detection of luciferase activity, which is the assay's readout, does not necessarily reflect cell viability? Hypothetically, and depending on the

mechanism of action, some active compounds could induce cells to release their contents to the extracellular space, which would presumably still allow for the detection of luciferase activity (and the dismissal of the active compound). Do the authors have any indication that this could be happening? For instance, is luciferase activity detected in the sporozoite media collected before cell lysis?

6. Please state the number of independent experiments in every figure.

7. Reference 14 is not correctly formatted.

Reviewer #2 (Remarks to the Author):

The manuscript by Miglianico et al., describes the validation of several sporozoite-stage assays to assess known antimalarial activities and as well as screen new compounds. The development of high-throughput assays to test the effects of compounds at several critical infection stages in detail is thorough and well done. A case can be made for employing the proposed screening cascade to identify potential novel antimalarials.

Major comments

The abstract could be more compelling and send a clearer message. The first result mentioned is that of the liver stage development assay, when the focus of the manuscript is on the less well studied sporozoite infection stages. More emphasis could be placed on how understudied the infection stages are, when it comes to drug discovery, both in the abstract and introduction. Also, a stronger use case could be made emphasizing the precedent of targeting sporozoites for vaccines (as mentioned), despite the very small time window the sporozoites are extracellular.

The first results section is a little confusing for Figure 1D. Is the x-axis time, the incubation period for the assay? The mention of incubation in Leibovitz medium seems strange, should it be 0.1% DMSO (all wells are incubated in Leibovitz medium)? The incubation period is the duration of the assay, correct? Not the amount of time sporozoites are incubated before compounds are added and the assay is run? The text states 1 μ M gramicidin while the figure legend states 10 μ M gramicidin.

Liver stage development assay: The transition to using strain NF175 is abrupt. Please explain in a sentence why to test it. The text says Figure S4 are results with strain NF175, while the Figure S4 legend says NF54. I'm assuming the miniaturization was feasible with NF175? I'm wondering why the Z' values are not reported for the 384-well assay developed? The methods should mention the two different liver stage development assays described. Would it not be better to have the NF175 results in the main Figure 2 and the NF54 96-well results in the Figure S4? It is not immediately clear when compound was added, please add this to the text and the methods.

It would be interesting to see a time course of how long gramicidin needs to be present with the sporozoite to inhibit schizogony. How short can pre-incubation with gramicidin be and still inhibit schizogony? As you mentioned compounds need to act quickly on sporozoites as they are only present in the blood for a very short period of time. The viability assay has an incubation period of 24 hours?, the motility 4 hours? The traversal assay for 1 hour? And the liver stage development assay for unknown period. Please discuss in more detail the incubation periods and how this translates to in vivo.

High-throughput screening of 2 libraries..: This section would read more clearly if the results were presented in the order of liver stage development (as the validation of the assays were presented). The focus is on the liver stage development assay because there are more actives, but this is expected

as you have stated that this stage is more studied and more similar to ASB stage. The lack of activity on the sporozoites in the other assays is the more interesting part. These stages may be harder to kill, but if there is an effect, it is lasting. Could you sort the table in Figure 3, by activity in the sporozoite viability assay? Then you would see the trend to having activity across all of the assays?

Discussion: I don't understand the sentiment in the first paragraph of the discussion. The cell traversal assay has all Z' values above 0.5, so I'm not sure why it's stated that it would need additional optimization. Furthermore, $Z' > 0.5$ is not usually necessary for a phenotypic screening-based assay such as this (more for biochemical assays). See reference PMID:22553881. The following sentences describes the advantages of the luciferase assay. This section needs re-working to have a more clear message about the advantages of the assays and what further work needs to be done.

Minor comments

Abstract: The first sentence primary should be moved to second part of the sentence to describe the target of experimental vaccines as sporozoites are the 'only' cause of infection of the vertebrate host.

The problem presented in the beginning of the abstract is that little is known about the sporozoites susceptibility to chemical intervention. Several assays were done, but then the abstract focuses on the hepatic activity, for which there have been numerous screens to assess efficacy of compounds.

Introduction: mention artemisinin resistance?

Line 40: 'the' number of cases and deaths has increased.

Line 214: 'chimeric'

Figure 1 legend: The 'F' callout should be 'E'. There is no 1E callout in the text. Please mention why the dotted line at 0.5 for Figure 1E.

Figures S1B, C are not mentioned in the text.

Figure 2 legend: please indicate what type of replicates are used for B,D,F,G, and I. What do each of the dots represent? A single well per plate, aggregation of control wells per plate?

Traversal and Liver stage development assays: please provide additional details of the modules and parameters used to quantify the images on the ImageXpress Pico.

Line 249: 0.1%

The readability of Figure 3 could be improved. There are not column headers for all of the columns. 'Value 1' is cutoff in all the columns. The color bar is not annotated (% inhibition?). The legend does not have the abbreviations in the figure defined.

Line 294: 'in this set'. Does the set refer to the whole 'experimental and marketed antimalarials set' or the compounds that were active in the liver stage development assay?

Table 1: where do the data for the ASB come from? If data was determined in-house, please add to methods and state in text. If data is from a reference, please cite it in the table.

Line 326: citation for reFRAME library

COMMSBIO-22-3415

Assessment of the drugability of the initial stages of Plasmodium falciparum malaria infection through miniaturized sporozoite assays and high throughput screening

Point to point response to the reviewers

Reviewer #1 (Remarks to the Author):

The manuscript by Miglianico, Bolscher and colleagues describes the development of a high-throughput screening pipeline to test the feasibility of targeting the sporozoite stage of P. falciparum through chemical interventions. This is an interesting concept that has only recently started to gain some traction, as nicely outlined in the Discussion. The authors established their screening pipeline by adapting well known assays, each measuring distinct sporozoite properties (gliding motility, cell traversal, hepatocyte invasion/development into exoerythrocytic forms), for high-content imaging, feature extraction and quantification. The manuscript also details the development of a new bioluminescence-based assay to measure sporozoite viability. As proof-of-concept, the authors screened two large compound libraries using the viability assay and an additional 78 compounds with known antimalarial activity using the entire pipeline. The output of these screening campaigns was somewhat underwhelming, as only a few hits show activity against the sporozoite stage. It is nevertheless reassuring that the single hit in the viability assay has been identified as an inhibitor of sporozoite motility in a previous study (reference 25). The authors conclude that the liver stage is more easily druggable than the sporozoite stage, which might indeed be the case and for several reasons, including, as they put forward, the unique proteome of this parasite stage. Is it possible that another reason might lie with differences in drug permeability through the membrane? Given the poor hit rates reported here, would future screens gain from using more focused compound libraries that tap into sporozoite-enriched biological pathways, such as cell motility?

Indeed, permeability may be an issue. We've added a sentence to the discussion to raise this (line 424). The idea of screening more focused libraries is very attractive, especially since the few successes in identifying sporozoite hits originate from screens monitoring motility, or a target based screen against MyoA that exerts an essential motor function. We adapted the discussion to include the suggestion of screening focused libraries (lines 476-478).

Overall, this is a well written manuscript that describes a screening approach designed to enable the identification of compounds offering chemoprotection against malaria. The authors should discuss this concept and provide references for chemoprevention strategies already being implemented in the field.

We have further expanded this concept in the introduction and added references (lines 50-64).

Specific comments:

1. Fig. 2 A shows the result of a gliding assay in which sporozoites were subject to a 4-hour pre-incubation at 4°C before the actual assay at 37°C. It is not clear why this pre-incubation step was necessary. The cell traversal assay, which has a similar duration (60 vs 90 min), was carried out

without this prolonged pre-incubation. Could the authors provide a rationale for why these two assays use a different drug incubation protocol?

Apologies, the Methods section described an exception to the standard protocol. The standard protocol for the gliding assay involves a 30' pre-incubation at room temperature, similar to the traversal assay. The experiments described in figure 2A were performed according to this standard protocol. Only the experiment represented in Figure 5C deviated from this protocol. The hits from the HTS campaign did not show any effect on sporozoite motility in initial experiments and we wondered whether a longer incubation time may reveal their activity. In order to preserve sporozoite motility during the pre-incubation, we needed to lower the temperature to 4 C for this experiment. We have now clarified this in the methods section (lines 184-187).

2. Do the authors observe any changes in sporozoite viability after the pre-incubation step at 4°C? Does it adversely affect sporozoite motility or their ability to infect hepatocytes?

The pre-incubation at 4°C was only applied to the gliding experiment depicted in Figure 5C. In the presence of serum sporozoites lose gliding ability after incubation for 4 hours at room temperature. Therefore the temperature for this prolong preincubation was lowered to 4 C. This had a minimal effect on gliding motility.

3. The switch to NF175 parasites in Fig. 2 G-I is somewhat confusing. Perhaps those panels could be moved to a Supplementary Figure, as they mainly reinforce previous reports that donor-dependent variation in infection is to be expected (<https://doi.org/10.15252/emboj.2020106583>).

The NF175 parasites were introduced at this point in the text as they are used in some the subsequent experiments given their better infection rates. We have modified the text to provide more introduction (lines 280-284) and clarified in the legends of the subsequent figures which parasite strain has been used. We have included a reference to previous work from our lab showing that drug sensitivities between NF54 and NF175 are identical (line 282). We've also included the reference suggested by the reviewer (line 284).

4. The authors show in Fig. 2 H-I that the hepatocyte permissiveness to infection is donor dependent and that the IC50 of primaquine also varies from donor to donor. In the screen of experimental and marketed antimalarials shown in Fig. 3, it is not clear if the hepatocytes used in each replicate originated from the same or different donors (please specify this point in the legend of both Fig. 3 and Fig. 4). Do the authors foresee the need to implement steps, such as pooling different hepatocyte donors or validate hits across donors, to account for this natural variability?

The experiments represented in Figures 3 and 4 used the same hepatocytes and we have clarified this in the legends.

The case of primaquine may be exceptional as this compound needs to be metabolically activated. The global portfolio malaria discovery portfolio puts a strong emphasis on metabolically stable compounds and the resulting compounds are likely less sensitive to variation in hepatocyte metabolism. Nevertheless, it may indeed be advisable to validate hits across donors, and we now mention this in the revised discussion (lines 432-437).

5. The hit rate in the viability assay was extremely low, and while the authors offer several good explanations for this outcome, is it possible that the detection of luciferase activity, which is the assay's readout, does not necessarily reflect cell viability? Hypothetically, and depending on the mechanism of action, some active compounds could induce cells to release their contents to the extracellular space, which would presumably still allow for the detection of luciferase activity (and the dismissal of the active compound). Do the authors have any indication that this could be happening? For instance, is luciferase activity detected in the sporozoite media collected before cell lysis?

We have no examples of such mechanism but, admittedly, did not test for it.

6. Please state the number of independent experiments in every figure.

Done.

7. Reference 14 is not correctly formatted.

Done.

Reviewer #2 (Remarks to the Author):

The manuscript by Miglianico et al., describes the validation of several sporozoite-stage assays to assess known antimalarial activities and as well as screen new compounds. The development of high-throughput assays to test the effects of compounds at several critical infection stages in detail is thorough and well done. A case can be made for employing the proposed screening cascade to identify potential novel antimalarials.

Major comments

The abstract could be more compelling and send a clearer message. The first result mentioned is that of the liver stage development assay, when the focus of the manuscript is on the less well studied sporozoite infection stages. More emphasis could be placed on how understudied the infection stages are, when it comes to drug discovery, both in the abstract and introduction. Also, a stronger use case could be made emphasizing the precedent of targeting sporozoites for vaccines (as mentioned), despite the very small time window the sporozoites are extracellular.

We modified the abstract to put less emphasis on the liver stage development assay. In the introduction we have highlighted that vaccination can provide sterile protection, despite the relatively short time the sporozoites are extracellular (lines 57-58).

The first results section is a little confusing for Figure 1D. Is the x-axis time, the incubation period for the assay? The mention of incubation in Leibovitz medium seems strange, should it be 0.1% DMSO (all wells are incubated in Leibovitz medium)? The incubation period is the duration of the assay, correct? Not the amount of time sporozoites are incubated before compounds are added and the assay is run? The text states 1 uM gramicidin while the figure legend states 10 uM gramicidin.

Apologies for the confusion. The x-axis indeed indicates incubation time. We compared vehicle control (0.1% DMSO in Leibovitz medium) to 1 μ M gramicidin in Leibovitz medium and have clarified this in the text (line 243).

Liver stage development assay: The transition to using strain NF175 is abrupt. Please explain in a sentence why to test it. The text says Figure S4 are results with strain NF175, while the Figure S4 legend says NF54. I'm assuming the miniaturization was feasible with NF175? I'm wondering why the Z' values are not reported for the 384-well assay developed? The methods should mention the two different liver stage development assays described. Would it not be better to have the NF175 results in the main Figure 2 and the NF54 96-well results in the Figure S4? It is not immediately clear when compound was added, please add this to the text and the methods.

We have inserted text to introduce NF175 (lines 280-284). Results in S4 were indeed with NF175, we have corrected this in the figure legend. We have also included a panel with Z' values from 21 independent experiments showing an average Z' of 0.36.

During the course of the project we switched from NF54 to NF175 parasites for liver schizogony assays, given the better infection rates for NF175. We have included a reference to our previous work showing drug sensitivities between the two lines are identical (line 282). We have clarified the use of specific strains in all legends. In addition, we have clarified the time compounds were added in the methods (lines 207-208) and legend to Figure 6.

It would be interesting to see a time course of how long gramicidin needs to be present with the sporozoite to inhibit schizogony. How short can pre-incubation with gramicidin be and still inhibit schizogony? As you mentioned compounds need to act quickly on sporozoites as they are only present in the blood for a very short period of time. The viability assay has an incubation period of 24 hours?, the motility 4 hours? The traversal assay for 1 hour? And the liver stage development assay for unknown period. Please discuss in more detail the incubation periods and how this translates to in vivo.

In the motility and traversal assays we have used a pre-incubation of sporozoites with compound of 30 minutes, followed by 90 and 60 minutes for the cells to display gliding motility and traversal, respectively. In the viability assay we have tested incubation times with gramicidin between 5 and 24 hours (Fig. 1E) and show that longer incubation times lead to a further drop in luciferase signal. Presumably the drop in luciferase activity is delayed relative to the actual death of the sporozoite, due to the half life of the luciferase enzyme. In the liver schizogony assay, gramicidin was active when sporozoites were incubated with the compound for 30' prior to hepatocyte infection but not when the compound was added after infection. We have further clarified the incubation times in the various assays in the revised manuscript (lines 149, 168-169, 193-194, 207-108).

High-throughput screening of 2 libraries..: This section would read more clearly if the results were presented in the order of liver stage development (as the validation of the assays were presented). The focus is on the liver stage development assay because there are more actives, but this is expected as you have stated that this stage is more studied and more similar to ASB stage. The lack of activity on the sporozoites in the other assays is the more interesting part. These stages may be harder to kill, but if there is an effect, it is lasting. Could you sort the table in Figure 3, by activity in the sporozoite viability assay? Then you would see the trend to having activity across all of the assays?

We have modified the order in the presentation of the compound set that was screened across all assays to highlight the sporozoite viability/motility and traversal assays (lines 303-304). In addition, we have sorted the results in table 1 on basis of the sporozoite viability assay.

Discussion: I don't understand the sentiment in the first paragraph of the discussion. The cell traversal assay has all Z' values above 0.5, so I'm not sure why it's stated that it would need additional optimization. Furthermore, Z' > 0.5 is not usually necessary for a phenotypic screening-based assay such as this (more for biochemical assays). See reference PMID:22553881. The following sentences describes the advantages of the luciferase assay. This section needs re-working to have a more clear message about the advantages of the assays and what further work needs to be done.

Indeed the Z' values of the traversal assays were above 0.5 whereas for the other assays these averaged below Z'. We agree that this may not be problematic for phenotypic (dose response) assays. The biggest bottleneck for high throughput screening is in generation of sufficient sporozoites. Here the viability assay has a clear advantage as it requires considerably less sporozoites in comparison with the other assays. We have further elaborated on this in the revised discussion (lines 410-417).

Minor comments

Abstract: The first sentence primary should be moved to second part of the sentence to describe the target of experimental vaccines as sporozoites are the 'only' cause of infection of the vertebrate host.

As sporozoites and not vaccines are the main subject of the paper we are a bit reluctant to make vaccines the subject of the first sentence but have modified is at follows:

"The sporozoite stages of malaria parasites are the primary cause of infection of the vertebrate host and are targeted by (experimental) vaccines."

The problem presented in the beginning of the abstract is that little is known about the sporozoites susceptibility to chemical intervention. Several assays were done, but then the abstract focuses on the hepatic activity, for which there have been numerous screens to assess efficacy of compounds.

We have de-emphasized the result from the hepatic assay in the abstract

Introduction: mention artemisinin resistance?

done (lines 39-40)

Line 40: 'the' number of cases and deaths has increased.

corrected

Line 214: 'chimeric'

corrected

Figure 1 legend: The 'F' callout should be 'E'. There is no 1E callout in the text. Please mention why the dotted line at 0.5 for Figure 1E.

Corrected. The dotted line indicates a Z' of 0.5, mentioned in the text, and we have clarified this in the legend.

Figures S1B, C are not mentioned in the text.

Corrected

Figure 2 legend: please indicate what type of replicates are used for B,D,F,G, and I. What do each of the dots represent? A single well per plate, aggregation of control wells per plate?

Done

Traversal and Liver stage development assays: please provide additional details of the modules and parameters used to quantify the images on the ImageXpress Pico.

We have inserted a more elaborate description in the methods section (lines 211-215)

Line 249: 0.1%

corrected

The readability of Figure 3 could be improved. There are not column headers for all of the columns. 'Value 1' is cutoff in all the columns. The color bar is not annotated (% inhibition?). The legend does not have the abbreviations in the figure defined.

corrected

Line 294: 'in this set'. Does the set refer to the whole 'experimental and marketed antimalarials set' or the compounds that were active in the liver stage development assay?

To the whole set, we have clarified this in the text (lines 318-319)

Table 1: where do the data for the ASB come from? If data was determined in-house, please add to methods and state in text. If data is from a reference, please cite it in the table.

We have inserted a reference in the table legend.

Line 326: citation for reFRAME library

Done (line 355)

REVIEWERS' COMMENTS:

Reviewer #1 (Remarks to the Author):

The authors have adequately addressed the comments raised by myself and the other reviewer. The revised manuscript has a clear message and reads very well.

Reviewer #2 (Remarks to the Author):

The authors have thoroughly addressed all comments by reviewers.